# Crystal Structure and Chemical Bonds in [Cu^II^_2_(Tolf)_4_(MeOH)_2_]∙2MeOH

**DOI:** 10.3390/ijms24021745

**Published:** 2023-01-16

**Authors:** Irena Majerz, Marta S. Krawczyk

**Affiliations:** Faculty of Pharmacy, Wrocław Medical University, Borowska 211a, 50-556 Wrocław, Poland

**Keywords:** tolfenamate ligand, paddle-wheel-like structure, copper coordination compound, QTAIM, NCI

## Abstract

A new coordination compound of copper(II) with a tolfenamate ligand of the paddle-wheel-like structure [Cu^II^_2_(Tolf)_4_(MeOH)_2_]∙2MeOH was obtained and structurally characterized. Chemical bonds of Cu(II)∙∙∙Cu(II) and Cu(II)–O were theoretically analyzed and compared with the results for selected similar structures from the CSD database. QTAIM analysis showed that the Cu(II)∙∙∙Cu(II) interaction has a strength comparable to a hydrogen bond, as indicated by the electron density at a critical point. The remaining QTAIM parameters indicate stability of the Cu(II)∙∙∙Cu(II) interaction. Other methods, such as NCI and NBO, also indicate a significant strength of this interaction. Thus, the Cu(II)∙∙∙Cu(II) interaction can be treated as one of the noncovalent interactions that affects the structure of the coordination compound, the packing of molecules in the crystal, and the general properties of the compound.

## 1. Introduction

Tolfenamic acid belongs to the group of fenamic acids (N-arylanthranilic acids), which are very popular compounds among the non-steroidal anti-inflammatory drugs (NSAIDs) that have long been used to treat inflammation, pain, and fever. Many NSAIDs exhibit chemopreventive and chemosuppressive effects [1,2,3,4,5] in different types of cancer [6,7]. The fenamic acids commonly used in pharmacy are mefenamic, tolfenamic, flufenamic, and meclofenamic acid. The clinical action and effectiveness of tolfenamic acid are known in the treatment of migraines [8], dysmenorrhea [9], as well as in rheumatic diseases [10,11]. All fenamic acids have undesired side effects; therefore, the simplest compound in this group—fenamic acid—is out of therapeutic application. Apart from side effects, the problem limiting the application of fenamic acids is their low water solubility, which a determinant of their low bioavailability. One of the best methods of improving the water solubility and bioavailability is the synthesis of fenamic acid salts with metal cations [12,13,14]. An additional advantage of using salts with metal cations is the supply of cations that play an important role in biological systems [15,16]. Among metal cations which can be used to produce salts or coordination compounds with organic drugs, Cu^2+^ plays an essential role due to its relatively low toxicity in comparison to other metal cation compounds [17,18], synergistic inflammatory properties [19], and potential anticancer properties [20,21,22]. 

The physical, biological, and therapeutic drug properties are related to the molecular structure; therefore, the crystal structure of a new drug is crucial in predicting and explaining properties, and determining the crystal structure is the first step in research leading to the design of a new drug.

In this paper we have undertaken determination of the crystal structure of a new dinuclear copper(II) coordination compound with tolfenamate ligands of the formula [Cu^II^_2_(Tolf)_4_(MeOH)_2_]∙2MeOH (Tolf = tolfenamate anion). The most characteristic feature of this structure is the paddle-wheel entity with the bond linking two Cu^2+^ cations and bonds linking the cations with the carboxylate groups. 

It is known that apart from typical covalent bonds, there are many noncovalent bonds, which, despite being weaker than covalent ones, very often determine the packing of molecules in the crystal, as well as their physicochemical properties. Commonly known noncovalent interactions that are fundamental in natural compounds are hydrogen bonds, ion–ion, van der Waals, and dispersive interaction [23] The range of typical noncovalent interactions known for a very long time has recently expanded to include pnictogen interactions [24,25,26]. Therefore, the question arises whether the interaction of Cu(II)∙∙∙Cu(II) is also one of the noncovalent interactions. An interesting approach seems to be to consider what type of bond Cu(II)∙∙∙Cu(II) is in coordination compounds of drugs, in terms of currently used theoretical methods, in order to broaden the knowledge of chemical bonds. Research into the nature of chemical bonds, in particular metal–metal bonds in multinuclear coordination compounds, may be crucial in the analysis of clusters and multinuclear compounds occurring in biological systems. Therefore, the search for answers to fundamental questions about the nature and strength of chemical bonds, including metal–metal bonds, using current theoretical methods may in the future contribute to the clarification of their properties and find correlations between structure and function.

On the basis of a Cambridge Structural Database (CSD, Version 5.43) [27] search, two other representative structures were selected for which the Cu^2+^∙∙∙Cu^2+^ and Cu^2+^–O bond lengths were also analyzed. The proposed Cu(II) paddle-wheel-like structures for which theoretical studies were performed are [Cu_2_(CH_3_COO)_4_(Hbet)_2_] (CSD refcode: QEBQIX) [28], the structure of a Cu(II) coordination compound with betaine, and [Cu_2_(CH_3_COO)_4_(CH_4_N_2_O)_2_]∙H_2_O (CSD refcode: ACURCU01) [29], a Cu(II) compound with urea. 

To analyze the character of copper–copper bonds, much theoretical research has been done. The first investigations for metal–metal bonds were carried out by Hoffmann and co-workers [30,31] and Cotton et al. [32]. Copper(I) dinuclear coordination compounds were extensively studied by Pyykkӧ and co-workers [33,34] and Carvajal et al. [35]. Recent research by Dinda and Samuelson presented the AIM analysis for Cu(I)∙∙∙Cu(I) bonds in copper(I) compounds [36].

In this work, the theoretical analysis for Cu(II)∙∙∙Cu(II) and Cu(II)–O bonds using the quantum theory of atoms in molecules (QTAIM) [37] has been carried out. According to this theory, two atoms are bonded if they are located at a common bond path with a bond critical point (BCP), which is a saddle point of the electron density on the gradient path. Electron density at the BCP is directly related to the bond strength [38,39], and the sign of the Laplacian of the electron density at the BCP indicates the interaction type. Concentration of electron density between atoms is connected with the negative sign of the Laplacian characteristic for the shared shell interaction—covalent and polarized bonds and depletion of the electron density at the BCP are connected to the positive sign of the Laplacian characteristic for the open-shell interaction, such as the hydrogen bond, van der Waals, and ionic interaction [40]. Stability of the bond is described by ellipticity (ε) of the electron clouds at the BCP [41,42] and the nonlinearity of the bond path, which cannot be too bent to avoid bond instability. The chemical bond and intermolecular interaction can be also characterized with the atomic net charge—q(A), the electron population of a particular atom N(A), %δ(A,B)—the average number of electrons in atom A that are delocalized between atom A and other atoms in the molecule, Bond(A,A’)/2—the number of electron pairs in atom A that are delocalized (shared) between atom A and atoms to which atom A is bonded, δ(A,B)—the electron delocalization index for atoms A and B, i.e., the average number of electrons delocalized (shared) between atoms A and B obtained by numerical integration of the charge density over the basin of the atom.

In the studied crystals, weak interactions were also analyzed by the Noncovalent Interaction (NCI) approach [43]. So far, the presented work is the first to discuss the nature of interactions between copper(II) ions in polynuclear coordination compounds using theoretical methods for characterizing noncovalent interactions, the QTAIM and NCI methods.

## 2. Results

To date, the Cambridge Structural Database [27] presents 1613 hits corresponding to the search for paddle-wheel-like double-core Cu(II) deposited structures with carboxylate ligands. The structures were searched for using the following criteria: the range for Cu(II)∙∙∙Cu(II) distance at 2.40 to 3.27 Å. Among this, 778 hits are the structures of compounds in which, in addition to carboxylates, there are also other ligands coordinating to copper(II) by an oxygen atom, while in 672 hits, non-carboxylate ligands are linked to Cu(II) via a nitrogen atom. These two groups of compounds containing, in addition to the carboxylate ligands, O- or N-donor ligands, respectively, differ somewhat in the length of the Cu(II)∙∙∙Cu(II) bonds. Compounds with O-donor ligands tend to form slightly shorter Cu(II)∙∙∙Cu(II) bonds compared to compounds with N-donor ligands, as shown by histograms generated in the CSD [27] (upper plot for structures with only O-donor ligands, bottom plot for structures with N-donor ligands in addition to carboxylates) (Figure 1, Appendix A).

In this paper, the crystal structure of a new paddle-wheel-like double-core Cu(II) coordination compound with tolfenamate and methanol ligands is presented, tetrakis(μ_2_-2-((3-Chloro-2-methylphenyl)amino)benzoato-O,O’)-bis(methanol)-dicopper(II)(Cu–Cu)–methanol(1/1) (Table 1). So far, the analogous structure of copper(II) compound with tolfenamate ligands and DMF molecules, tetrakis(μ_2_-2-((3-Chloro-2-methylphenyl)amino)benzoato- O,O’)-bis(N,N-dimethylformamide-O)-dicopper(II)(Cu–Cu) has been reported previously (CSD refcode: ARALUA) [44]. According to the data, fourteen paddle-wheel-like structures with fenamate ligands have been published [44,45,46,47,48,49,50,51,52]. The Cu(II)∙∙∙Cu(II) bond length for the studied [Cu^II^_2_(Tolf)_4_(MeOH)_2_]∙2MeOH equals 2.5880(4) Å and is in the range of bond lengths typical for this group of compounds, ranging between 2.572 Å (for EDUBOX [45] —a compound with flufenamate and water ligands) and 2.627 Å (for SUTPIG [46]—a compound with mefenamate and DMSO ligands). For the aforementioned compound with tolfenamate and DMF ligands (ARALUA [44]), the copper(II)∙∙∙copper(II) bond equals 2.6074(18) A. In the case of structures coordinated with fenamates and ethanol or methanol molecules, the bond lengths are about 2.58 Å (POMHOP [48], EDUBUD [45]) and 2.59 Å (MADTES [49]).

Moreover, in the Cambridge Structural Database, there are also two paddle-wheel structures with a fenamate ligand and N-coordinated ligands as well, viz., with vinylpyridine (DIFQUG [53]) and with 4-methylpyridin-2-amine (LIFBAF [54]), for which the Cu(II)∙∙∙Cu(II) bond lengths equal 2.63 and 2.64 Å, respectively; this is in line with the observed trends in Figure 1. In our studies we compared the theoretical analysis for [Cu^II^_2_(Tolf)_4_(MeOH)_2_]∙2MeOH with the results obtained for two selected structures from the CSD database of [Cu_2_(CH_3_COO)_4_(Hbet)_2_] (Hbet = betaine) (QEBQIX) [28] and [Cu_2_(CH_3_COO)_4_(CH_4_N_2_O)_2_]∙H_2_O (CH_4_N_2_O_2_—urea) (CSD refcode ACURCU01) [29]. When selecting structures, we took into account low R-factors, the simplicity of structures with a low molecular weight, as well as the Cu(II)∙∙∙Cu(II) bond lengths from the typical range for paddle-wheel-like structures with O-donor ligands and from the range of the largest distances. For QEBQIX, the Cu^2+^∙∙∙Cu^2+^ distance equals 2.6897(7) Å, and for ACURCU01, it is 2.624(1) Å; these distances are significantly longer compared to the investigated compound (2.5844(4) Å).

### 2.1. Crystal Structure of [Cu^II^_2_(Tolf)_4_(MeOH)_2_]∙2MeOH

The studied compound tetrakis(μ_2_-2-((3-chloro-2-methylphenyl)amino)benzoate -O,O’)-bis(methanol)-dicopper(II)(Cu–Cu)–methanol(1/1) of the formula [Cu^II^_2_(Tolf)_4_(MeOH)_2_]∙2MeOH (Tolf = tolfenamate anion) is a dinuclear copper(II) coordination compound. The compound is composed of four tolfenamate anions and two methanol molecules coordinating to two copper(II) ions (Figure 2). The compound adopts the geometry of a so-called “paddle-wheel structure” in which each of the tolfenamate ions coordinates to both copper(II) centers in dimer, and each of the oxygen atoms of the carboxylate group is bonded to one copper(II) cation. The coordination sphere of each of the Cu^2+^ ions is a distorted octahedron. The interplanar angle between carboxylate groups is 89.59(4) Å. The studied compound is centrosymmetric.

Selected bond lengths and angles are presented in Table 2 and Table 3. The structure reveals the characteristic for fenamate’s intramolecular hydrogen bonds: N1A–H1A∙∙∙O2A and N1B–H1B∙∙∙O2B (Table 4). The intermolecular hydrogen bonds with methanol molecules are observed in which the oxygen atom from the uncoordinated methanol molecule (O2M) is an acceptor (O1M–H1M∙∙∙O2M) and a proton donor (O2M–H2M∙∙∙O1B^i^, i = −x + 1, −y + 1, −z + 1) as well (Table 4). The crystal is stabilized by C–H···π and Cl···π interactions. The adjacent molecules of [Cu^II^_2_(Tolf)_4_(MeOH)_2_] interact via C11B–H11B···Cg1^ii^ and C10B–Cl2···Cg2^ii^ forming chains (ii = 1 −x, −y, 2 −z; the centers of the aromatic rings are denoted as Cg1 and Cg2; Cg1 = [C2A/C3A/C4A/C5A/C6A/C7A], Cg2 = [C2B/C3B/C4B/C5B/C6B/C7B]) (Figure 3, Appendix A). Interactions of [Cu^II^_2_(Tolf)_4_(MeOH)_2_] molecules with uncoordinated methanol molecules through C–H···π (C2M–H2MC···Cg3, Cg3 = [C8A/C9A/C10A/C11A/C12A/C13A]) are also observed (Appendix A).

### 2.2. QTAIM Analysis of [Cu^II^_2_(Tolf)_4_(MeOH)_2_]∙2MeOH, QEBQIX and ACURCU01

The nature of the interaction between metal ions has long been studied by theoretical methods [30,31,32,33,34,35,36]. The use of the QTAIM analysis in the study of the interaction of Cu^+^ ions in model compounds [36] allowed for the conclusion that, contrary to expectations, this interaction is not repulsive but is a weak closed-shell interaction. In order to analyze the structure of the investigated compound, in particular the bond between two copper ions, the results of the QTAIM analysis for the investigated compound have been compared with the results for the selected previously determined structures of ACURCU01 and QEBQIX, for which the Cu^2+^∙∙∙Cu^2+^ distances are 2.5844(4), 2.624(1), and 2.6897(7) Å, respectively. QTAIM diagrams for the tolfenamic acid Cu(II) complex, QEBQIX, and ACURCU01 are presented in Figure 4 and Appendix A.

QTAIM parameters describing the selected covalent chemical bond and weak interaction for Cu^2+^∙∙∙Cu^2+^ and Cu^2+^∙∙∙O for all the investigated compounds are collected in Table 5. In Table 6, the atom basin parameters are gathered. The parameter which allows differentiation of the open-shell and closed-shell interaction is the sign of Laplacian of the electron density. Depletion of the electron density between the atoms gives a positive sign of the Laplacian, which is typical for hydrogen bonds, van der Waals, and ionic interactions. According to the values in Table 5, the Cu^2+^∙∙∙Cu^2+^ and Cu^2+^∙∙∙O bonds are the closed-shell interactions when the C–O and C–C are the open-shell interactions typical for the covalent bonds. Taking into account only the sign of Laplacian, it is difficult to distinguish whether the interaction between the Cu^2+^ centers and between Cu^2+^ and carboxylate oxygen atoms can be classified as a weak interaction or an interaction between ions. The main parameter reflecting the chemical bond and the weak interaction strength is the electron density ρ(r) at the BCP located between the interacting atoms. Comparison of electron density at the Cu^2+^∙∙∙Cu^2+^ BCP for the investigated coordination compounds, QEBQIX and ACURCUO01, shows that for a greater distance of Cu^2+^∙∙∙Cu^2+^, the interaction between the cations is weaker, and the changes in electron density are sensitive to the changes in the distance between the Cu^2+^ ions. Comparison of the electron density at the critical point between Cu^2+^∙∙∙Cu^2+^ ions and at the critical point of a typical covalent bond shows how weak this interaction is. 

Among the bonds listed in Table 5, the C–O bonds are characterized by the highest electron density at the BCP, which confirms the strength of the bond. Partially double-bond character of the C–O bonds is confirmed by the average number of the electrons delocalized between C and O atoms, which is higher than 1. For the single C–C bond, it is about 0.9. In contrast to the carboxylic group with single and double C–O bond, the geometrical and QTAIM parameters for both C–O bonds in every carboxylate group are similar, so both C–O bonds are identical with partially double character. The covalent C–C bond in tolfenamic acid is significantly weaker than the C–O bond. The closed-shell Cu^2+^∙∙∙O interaction between the Cu^2+^ cation and the tolfenamate carboxylate group is stronger than the interaction with the oxygen of methanol.

In line with the electron density is the delocalization index δ(A,B). The average number of electrons participating in the C–C bond is about 0.8. For the C–O bond, it is above 1, for Cu^2+^∙∙∙O about 0.4, and for the Cu^2+^∙∙∙O other than the carboxylate group, it is about 0.2. The delocalization index shows that the average number of electrons delocalized between Cu^2+^ cations is higher than between Cu^2+^∙∙∙O, but the electron density at the BCP indicates that the Cu^2+^∙∙∙O interaction is stronger. The other parameters in Table 5 confirm stability of every closed-shell interaction for the investigated compounds. Low ellipticity and linearity of the bonds and weak interactions suggest high stability of the complex.

Atomic properties are obtained by integration of three-dimensional electron density over the basin of the atom defined as the electron density local zero flux surfaces. The average electron population of atom N(A) can be obtained by numerical integration of the charge density over the basin of the atom. The net charge of atom q(A) is obtained by subtracting N(A) from the nuclear charge, q(A) = Z_n_ − N(A) [55]. The results of the integration over the atomic basin are presented in Table 6. The open-shell interactions are connected with a higher percentage of the electrons for atoms that are engaged in a covalent bond. δBond(A,A’)/2 is the number of electrons in atom A that are delocalized (shared) between atom A and atoms to which atom A is bonded. It is characteristic that in the case of the Cu^2+^ cations, only about 5% of electrons participate in the weak interaction linking the cations together, when for the atoms participating in a covalent bond, the amount of electrons is significantly higher.

### 2.3. NCI Analysis of the Weak Interaction for the [Cu^II^_2_(Tolf)_4_(MeOH)_2_]∙2MeOH and for QEBQIX

The second theoretical method, complementary to the QTAIM, which is very useful for characterizing the weak interaction, is the Noncovalent Interaction (NCI) approach [43]. This method allows differentiation of the repulsive and attractive interactions by using the reduced electron density gradient of the electron density, which describes the deviation from a homologous electron density distribution s = 1/(2(3*π*^2^)^1/3^)|∇ρ|/ρ^4/3^). The reduced gradient of electron density is very high for weak interactions but approaches zero for covalent bonds. The plot of the reduced electron density gradient versus the electron density multiplied by the sign of the second Hessian eigenvalue (λ_2_) of electron density makes it possible to differentiate a repulsive and attractive interaction. The spikes on the NCI diagram on the negative side of the vertical axis represent the attractive interactions, while those on the positive represent the repulsive interactions. Two symmetric peaks around zero represent the dispersive interaction. In addition to the NCI diagram, this method also offers a visualization of the interaction in the plots of the reduced density gradient in the real space for the molecule in colors traditionally used in the NCI approach: blue for attractive, red for repulsive, and green for intermediate-strength interactions (Figure 5, Appendix A).

According to the NCI method, the interaction between Cu^2+^ ions is dispersive rather than repulsive, which is consistent with the low electron density at the BCP between the cations. The strongest attractive interaction is for the Cu^2+^ cation and the methanol oxygen. The comparison of the Cu(II)∙∙∙O and Cu(II)∙∙∙Cu(II) interactions indicates a similar strength of these attractive interactions. A repulsive interaction is connected with the oxygen atoms of the carboxylate groups. 

### 2.4. NBO, HOMO—LUMO and Fukui Parameters

Formation of the covalent bonds and the weak interaction are connected to the reorganization of the electron cloud which, other than the QTAIM analysis, can be described in a traditional way represented by the NBO analysis, which refers to a traditional Löwdin structure [56]. While QTAIM is based on the electron density, the NBO analysis refers to a traditional description of the molecular orbitals. The QTAIM and NBO approaches are complementary and strongly related to each other [57,58].

According to the NLMO analysis, the interaction between Cu^2+^ cations is formed by the lone pairs of both cations (Figure 6). Each Cu^2+^ cation has four lone pairs, and only one of them is directed to the lone pair of the other cation. For the Cu^2+^ cation, the weak interaction is formed by the lone pair which consists of 98.97% of the d orbital, 0.13% of p, and 0.9% of s. Participation of the orbital of the second Cu^2+^ is 0.303%, and the orbital consists of 83.98% of s, 11.25% of p, and 4.31% of d. Interaction of the Cu^2+^ cation with carboxylate oxygen atoms is also connected with overlapping of the lone pairs directed to the cation. In the interaction of oxygen, it participates with two Cu^2+^ cations in similar percentages (4.09 and 5.593%). The oxygen orbital that forms the interaction is built of the s orbital (13.88%) and p orbital (86.02%). As the interaction covers two cations, the orbital is localized only in 86.07%, which confirms delocalization of the dispersive interaction between the Cu^2+^ cations.

HOMO and LUMO orbitals for Cu^II^_2_(Tolf)_4_(MeOH)_2_]∙2MeOH, QEBQIX, and ACURCU01 are presented in Figure 7. It is characteristic that for every investigated compound, the frontier orbitals are located in the center of the molecule and spread over the Cu^2+^ cations and the carboxylic groups. The HOMO-LUMO energy gap is similar for the investigated compounds and equals 33.13, 32.94, and 31.69 kcal/mol for Cu^II^_2_(Tolf)_4_(MeOH)_2_]∙2MeOH, QEBQIX, and ACURCU01, respectively.

The 3D model of molecular electrostatic potential is presented in Figure 8. The red color indicates the negatively charged oxygen atoms, while the blue represents positively charged Cu^2+^ cations. For QEBQIX, the positive and negative charge represents the betaine ligand.

To investigate reactivity of the paddle-wheel compounds for QEBQIX and ACURCU01, Fukui indices have been calculated and presented in Figure 9. Fukui indices provide information regarding which part of the molecule can lose or accept an electron, that is, whether it may undergo nucleophilic or electrophilic attack [59]. Because the added electrons go into the HOMO orbital and the removed electrons come from LUMO, Fukui indices describe the change of electron density in a frontier orbital, as a result of adding or removing an electron from the frontier orbitals.

The Fukui minus and Fukui plus shown in Figure 9 indicate that the place of nucleophilic and electrophilic attack is located in the central part of the molecule containing Cu^2+^ cations and oxygens from carboxyl groups. On the other hand, the isosurface value of 0.001 indicates that this part of the molecule is not highly reactive.

### 2.5. Delocalization of Electrons

Delocalization of the electron density, except for the distribution of electron density in the molecule, can be visualized in the space of the molecule and indicates the bonds with mobile electrons. Electron delocalization can be visualized with the ACID (Anisotropy of the Current-Induced Density) method [60], which should be used to complement electron density analysis (Figure 10).

According to the geometry and electron density parameters, the electrons for CO bonds with partially double character are characterized by high mobility identical to the aromatic rings. The C–C bond linking the carboxylate group with the aromatic ring is also characterized by high mobility. High mobility of the electrons is also characteristic of Cu^2+^ ions, but not for the weak interactions involving cations.

### 2.6. Decomposition of the Bonding Energy

The QTAIM and NCI methods illustrate the strength of the interactions. The NBO method explains the mechanism of orbital interaction. The energy decomposition according to Morokuma—Ziegler has been performed [61,62] and is an additional source of the interpretation of the bonding force linking the Cu^2+^∙∙∙Cu^2+^ cations with the surrounding atoms. The bonding energy is decomposed into electrostatic (E_elect_), Pauli (E_Pauli_), orbital (E_orb_), and dispersive (E_dysp_) components.
E_bonding_ = E_elect_ + E_Pauli_ + E_orb_ +E_dysp_
where the E_elect_ is the Coulomb interaction between the unperturbed charge of the two interacting fragments, E_Pauli_ expresses the destabilizing Pauli repulsion between the occupied orbitals, E_orb_ illustrates the interaction energy between the orbitals of both complex components, and E_dysp_ is the dispersion energy for the intermolecular interaction. The sum of E_elect_ and E_Pauli_ represents the steric interaction. The interaction energy has been calculated for two interacting fragments: the Cu^2+^∙∙∙Cu^2+^ core and the surrounding molecules. The calculated energy components in kcal/mol are as follows: E_elect_ −1679.20, E_Pauli_ 292.94, E_Steric_ −1386.26, E_dysp_ −13.58, E_orb_ −894.20. Total bonding energy is equal −2294.04. Energy decomposition shows that, despite the fact that the formation of the interactions with Cu^2+^ cations is connected with overlapping of the lone pairs, the complex is stabilized by the electrostatic interaction of the Cu^2+^∙∙∙ Cu^2+^ core and the surrounding molecules, although the orbital interaction is also significant. Dispersive interaction is important for the Cu^2+^∙∙∙Cu^2+^ cation interaction. 

## 3. Materials and Methods

### 3.1. Single-Crystal X-ray Diffraction

Crystals of [Cu^II^_2_(Tolf)_4_(MeOH)_2_]∙2MeOH coordination compound were measured with Xcalibur Ruby four-circle diffractometer equipped with a CCD area detector and a graphite monochromator using Mo Kα (λ = 0.71073 Å) radiation. The collected diffraction data were processed with the CrysAlis PRO program [63]. The structure was solved by the Patterson method and refined by the full-matrix least-squares method using SHELXT software [64,65]. The crystal data and structure refinement for the investigated compound are presented in Table 1. The program DIAMOND [66] was used for molecular graphics.

### 3.2. DTA, DTG

DTA (Differential Thermal Analysis) and DTG (Differential Thermal Gravimetry) were carried out by means of a Seteram SETSYS 16/18 instrument in the temperature range 330–870 K on a heating run at the rate of 5 K/min under N_2_. 

### 3.3. IR, Raman

IR studies in the range of 400–4000 cm^−1^ were carried out using Thermo Scientific USA model Nicolet iS50 Fourier Transform Infrared Spectrometer FTIR using ATR. Raman spectrum was collected using Bruker Bravo Raman spectrometer.

### 3.4. Theoretical Analysis

The wave function for the experimental structure was obtained with Gaussian16 software [67]. The QTAIM analysis was carried with the AIMALL program [68]. Weak noncovalent interactions were investigated with the NCI program [43]. To describe the delocalization of the electrons, the ACID program [60] was used. NBO, molecular orbital analysis, and partition energy were performed without the molecule optimization using the ADF program [69].

### 3.5. Materials and Synthetic Procedures

Crystals of [Cu^II^_2_(Tolf)_4_(MeOH)_2_]∙2MeOH coordination compound were obtained according to the analogous procedure for the synthesis of a copper coordination compound with meclofenamate ligand [17]. Copper(II) acetate monohydrate ([Cu(CH_3_COO)_2_(H_2_O)]_2_) (0.0250 g, 0.125 mmol) was dissolved in methanol (small amount), and the solution was added to a solution of tolfenamic acid (0.0654 g, 0.250 mmol) in methanol (about 5 mL). According to the preparative methods for the coordination compound of Cu(II) with meclofenamic acid, a few drops of triethylamine were added to the reaction mixture until the pH value was about 7. The reaction mixture was stirred for 2 h at room temperature, and then it was cooled in the refrigerator overnight. Slow evaporation of the solvent resulted in needle-like green crystals. The yield was fair (above 50%). M.p. 132–134 °C. IR (cm^−1^): 3330 w, ν(NH); 3078 vw; 2952 vw; 2833 vw; 1622 m, ν_asym_(COO^-^); 1586 s; 1565 m; 1512 s; 1459 s; 1447 s; 1392 vs, ν_sym_(COO^-^); 1323 vw; 1286 s; 1230 vw; 1208 vw; 1181 vw; 1164 m; 1152 m; 1122 vw; 1098 vw; 1079 vw; 1054 vw; 1044 vw; 1014 s; 957 vw; 912 w; 850 w; 805 w; 777 s; 751 vs; 736 w; 708 m; 680 s; 617 w. Raman (cm−1): 3160 w, 3130 w, 3078 w, 1980 w, 1951 w, 1915 w, 1858 w, 1611 vs, 1580 s, 1461 w, 1384 s, 1319 w, 1280 s, 1226 w, 1204 w, 1158 s, 1075 m, 1047 m, 849 s, 685 vw, 616 m, 520 m. Anal. calc. for C_58_H_56_N_4_O_10_Cl_4_Cu_2_ (coordination compound without uncoordinated methanol molecules) (1238 g mol^−1^): C, 56.3; H, 4.5; N, 4.5; O, 12.9; Cl, 11.5; Cu, 10.3; found: C, 56.8; H, 4.3; N, 4.4%.

The IR spectrum (Appendix A) exhibits the weak bands at 3330 (w) cm^−1^ derived from ν(NH) and at 3078 (vw) cm^−1^ attributed to ν(CH) from the aromatic ring. Bands at 2952 (vw) and 2833 (vw) cm^−1^ can be assigned to ν_asym_(CH_3_) and ν_sym_(CH_3_), respectively. The infrared spectrum for the studied crystal exhibits characteristic bands observed at 1622 (m) and 1392 (vs) cm^−1^ that can be assigned to ν_asym_(COO^-^) and ν_sym_(COO^-^) according to Hurtado et al. [70] (the band at 1392 (vs) cm^−1^ is broad; therefore, a composition of vibrations of ν_sym_(COO^-^) and δ_as_(CH_3_) [71] can occur). Bands at 1586 (s), 1565 (m), and 1512 (s) cm^−1^ can be attributed to a composition of ν(CN), bending vibrations of δ(NH) [71], and stretching vibrations of aromatic rings of ν(CC) [72], which correspond to analogous vibrations observed in the tolfenamic acid. The bands at 1459 (s) and 1447 (s) cm^−1^ could be assigned to δ(CH) as well as ν(CC) from aromatic rings. In the region of 1000–1300 cm^−1^, the δ(CH) from the aromatic ring appear, but the 1286 (s) cm^−1^ can also be assigned to ν(CN) vibration. According to Jabeen et al. [71], the strong band observed at 1014 (s) cm^−1^ can be attributed to ν(CCl) as well as ρ(CH_3_). In the region of 1000–675, bands observed can be assigned to δ(CH). Bands observed between 400–540 cm^−1^ may be interpreted as derived from ν(CCl) and δ(CCl) [71,72].

In the Raman spectrum (Appendix A), bands at about 3160, 3130, and 3078 cm^−1^ could be assigned to ν(CH) from aromatic rings. Broad weak bands with a maximum at about 2900 cm^−1^ can be attributed to ν_asym_(CH_3_) and ν_sym_(CH_3_) vibrations. Bands at about 1611 (vs) and 1580 (s) cm^−1^ can be assigned to δ(NH) and ν(CN), respectively. At 1461 (w) cm^−1^, the deformation modes δ(CH) can be expected. The strong band at 1384 (s) cm^−1^ can be interpreted as derived from δ(CH_3_). The bands observed at 1319 (w), 1280 (s) cm^−1^ can be attributed to ν(CN), and those at 1226 (w), 1204 (w), 1158 (s), 1075 (m), 1047 (m), 849 (s) cm^−1^ can be associated with ring deformations δ(CH) and δ(CC). At 685 (vw) cm^−1^, the deformation modes δ(NH) can be assigned. Bands observed at about 616 (m), 520 (m) cm^−1^ may be interpreted as derived from δ(CCl) [71]. 

Results of DTA (Differential Thermal Analysis) and DTG (Differential Thermal Gravimetry) studies are presented in Appendix A. DTA and DTG curves prove chemical and thermal stability of the studied compound up to about 330 K. Between 330 and 460 K, a distinct weight loss is observed, which is connected to the loss of methanol molecules present in the crystal as well as coordinated to copper(II) ions. Above 465 K, rapid weight loss attributed to further decomposition of the studied compound is observed. 

## 4. Conclusions

From the point of view of potential applications of double-core copper complexes with pharmaceutical ligands as new, promising therapeutic substances, it seems important to study the structure of compounds, the nature of chemical bonds, and weak interactions in the solid. Studying the nature of chemical bonds and interactions in the crystal may in the future be crucial in interpreting the mechanisms of drug delivery to cells and binding to proteins.The interactions linking the Cu^2+^∙∙∙Cu^2+^ cations as well as the core of the compound and the surrounding molecules are a weak, closed shell but very stable. Taking into account the value of the electron density at the critical point and the stability of the attracting Cu^2+^∙∙∙Cu^2+^ interaction, this interaction can be considered as one of the noncovalent interactions, affecting the overall geometry of the compound, the spatial arrangement of molecules in the crystal, and thus physicochemical properties, similar to the hydrogen bond, van der Waals, or pnictogen interaction.The interactions are formed by the orbital overlapping. The stabilizing force for the compounds is the electrostatic interaction of the Cu^2+^∙∙∙Cu^2+^ core with the rest of the molecule, especially with the carboxylate oxygen atoms.

## Figures and Tables

**Figure 1 ijms-24-01745-f001:**
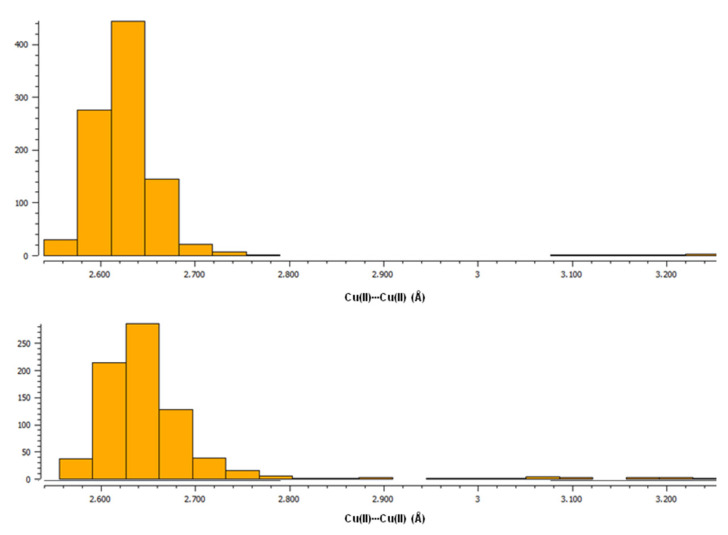
Histograms from CSD [27] showing the tendency of Cu(II)∙∙∙Cu(II) bond lengths in paddle-wheel-like double-core Cu(II) compounds with carboxylate ligands. (**Top**): a plot for structures with only O-donor ligands; (**Bottom**): a plot for structures with N-donor ligands apart from carboxylates.

**Figure 2 ijms-24-01745-f002:**
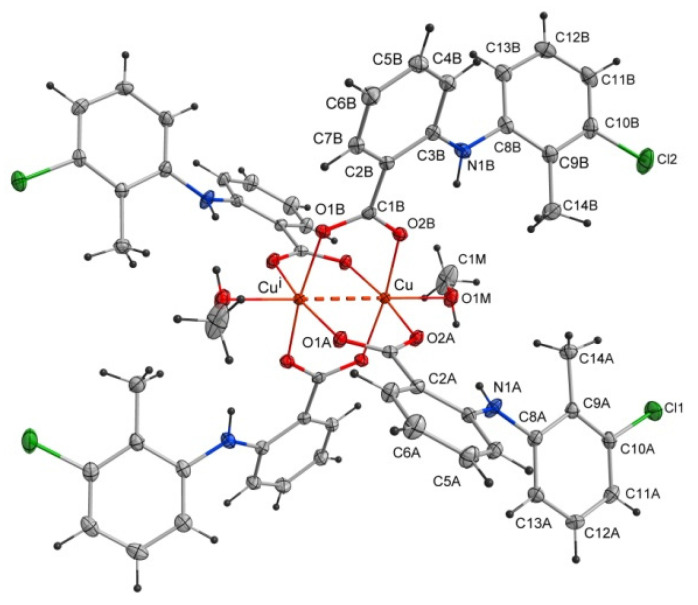
Structure of the coordination compound of [Cu^II^_2_(Tolf)_4_(MeOH)_2_]∙2MeOH. For clarity, uncoordinated methanol molecules were removed. The dashed line shows the distance between copper(II) ions. Symmetry code: i = −x + 1, −y + 1, −z + 1.

**Figure 3 ijms-24-01745-f003:**
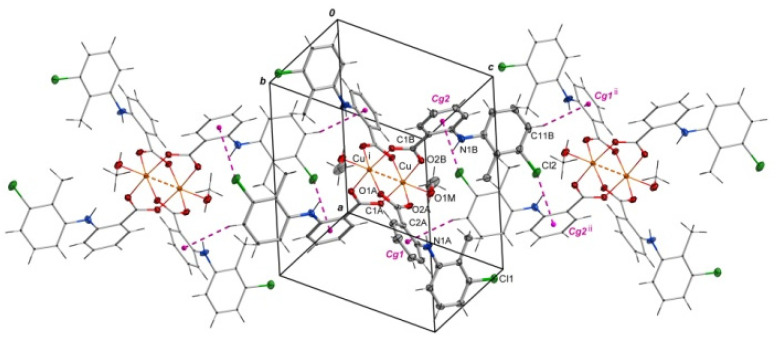
Arrangement of molecules of [Cu^II^_2_(Tolf)_4_(MeOH)_2_] in a chain where adjacent molecules contact via Cl···π and C–H···π interactions. For clarity, uncoordinated methanol molecules were removed. Symmetry codes: i = −x + 1, −y + 1, −z + 1; ii = 1 −x, −y, 2 −z.

**Figure 4 ijms-24-01745-f004:**
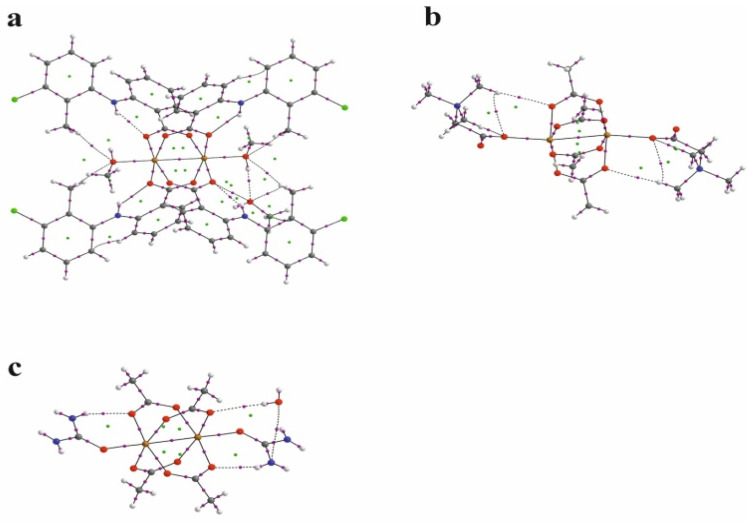
QTAIM diagrams for [Cu^II^_2_(Tolf)_4_(MeOH)_2_]∙2MeOH (**a**), QEBQIX (**b**), and ACURCU01 (**c**). The lines connecting the nuclei are the bond paths. Red and yellow dots represent the BCPs and RCPs, respectively.

**Figure 5 ijms-24-01745-f005:**
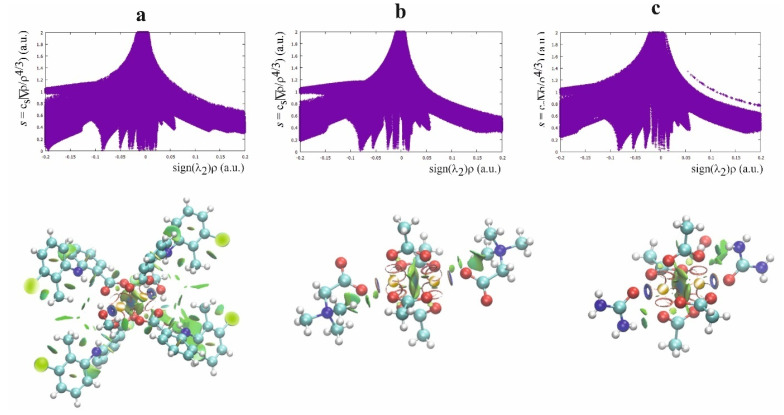
NCI diagram and NCI plot for [Cu^II^_2_(Tolf)_4_(MeOH)_2_]∙2MeOH (**a**), QEBQIX (**b**), and ACURCU01 (**c**).

**Figure 6 ijms-24-01745-f006:**
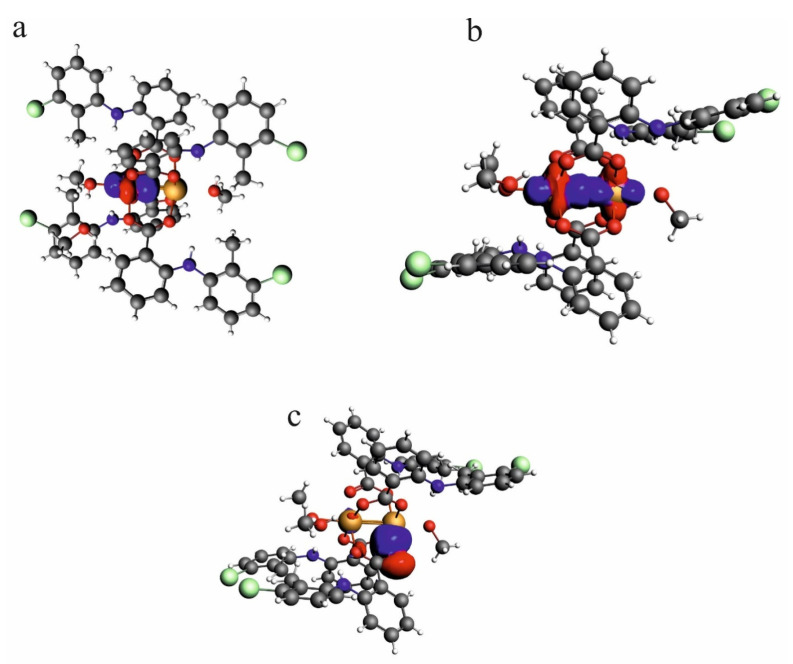
NLMO orbitals for lone pair on one of the Cu^2+^ cations in the investigated compound—[Cu^II^_2_(Tolf)_4_(MeOH)_2_]∙2MeOH (**a**), the weak interaction formed by lone pairs of two Cu^2+^ cations (**b**), lone pairs on the oxygen atom which is engaged in the interaction with Cu^2+^ cation (**c**).

**Figure 7 ijms-24-01745-f007:**
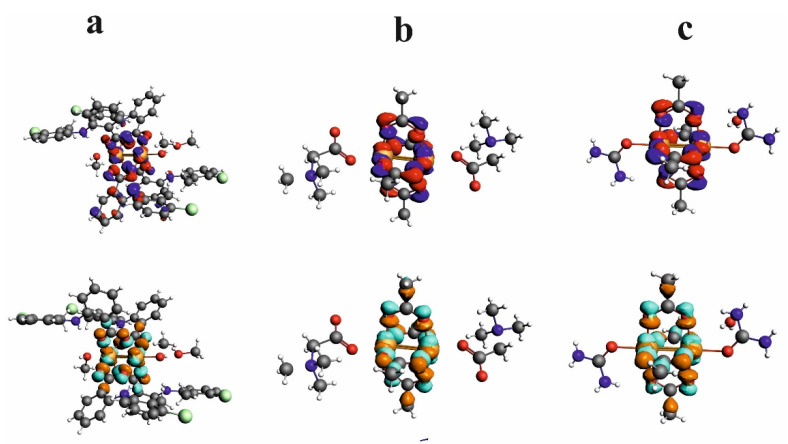
HOMO (**upper row**) and LUMO (**bottom row**) orbitals for [Cu^II^_2_(Tolf)_4_(MeOH)_2_]∙2MeOH (**a**), QEBQIX (**b**), and ACURCU01 (**c**).

**Figure 8 ijms-24-01745-f008:**
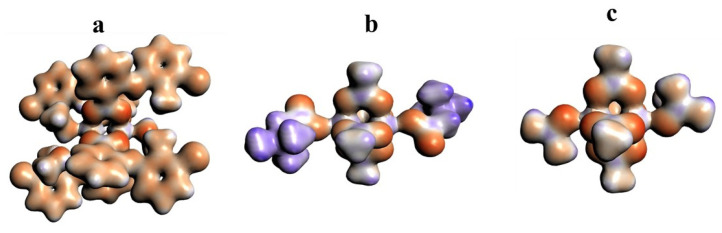
Electrostatic potential surfaces for [Cu^II^_2_(Tolf)_4_(MeOH)_2_]∙2MeOH (**a**), QEBQIX (**b**), and ACURCU01 (**c**). Field isosurfaces have been generated at the value of 0.03. Red—negative, blue—positive.

**Figure 9 ijms-24-01745-f009:**
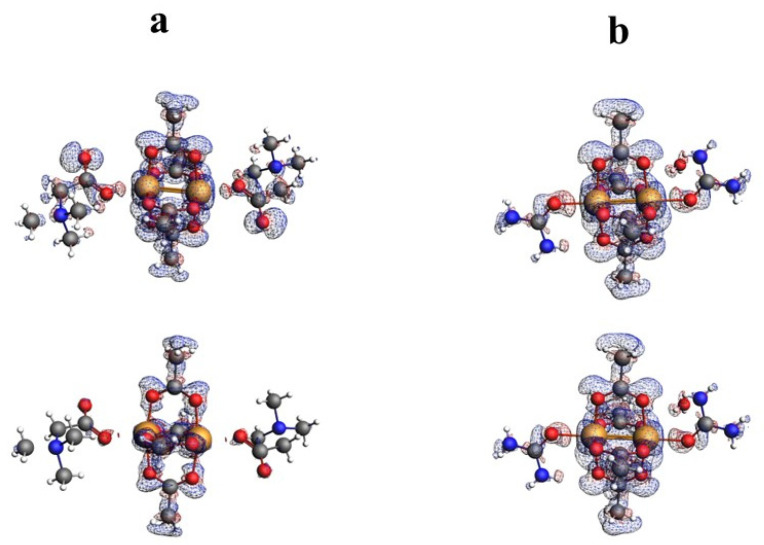
Fukui surfaces for QEBQIX (**a**) and ACURCU01 (**b**). Field isosurfaces have been generated at the value of 0.001. Upper row—Fukui minus, bottom row—Fukui plus.

**Figure 10 ijms-24-01745-f010:**
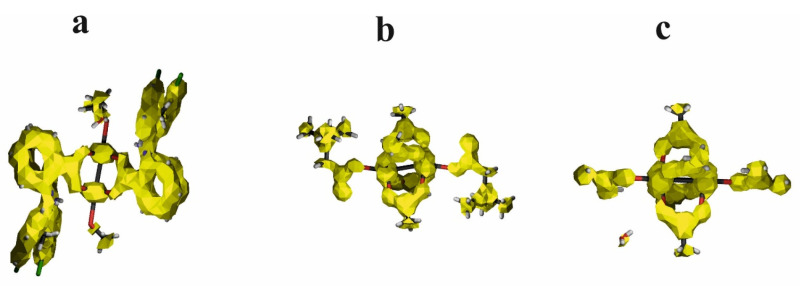
ACID surfaces for [Cu^II^_2_(Tolf)_4_(MeOH)_2_]∙2MeOH (**a**), QEBQIX (**b**), and ACURCU01 (**c**). ACID field isosurfaces have been generated at the value of 0.05.

**Table 1 ijms-24-01745-t001:** Crystallographic data and refinement parameters for [Cu^II^_2_(Tolf)_4_(MeOH)_2_]∙2MeOH.

Crystal Data	
Chemical formula	[Cu_2_(C_14_H_11_ClNO_2_)_4_(CH_3_OH)_2_]·2(CH_3_OH)
M_r_	1298.00
Crystal system, space group	Triclinic, *P*$\bar{1}$
Temperature (K)	100(2)
*a* (Å)	11.3917(3)
*b* (Å)	11.7297(4)
*c* (Å)	12.1260(4)
*α* (^o^)	68.20(3)
*β* (^o^)	77.83(3)
*γ* (^o^)	81.46(3)
*V* (Å^3^)	1466.2(4)
*Z*	1
Radiation type	Mo *K*α
Crystal size (mm)	0.64 × 0.11 × 0.08
*Data collection*	
Diffractometer	Xcalibur Ruby-CCD κ-geometry diffractometer
No. of measured, independent, and observed [I > 2*σ*(*I*)] reflections	13,769, 8096, 6803
*R_int_*	0.026
*Refinement*	
*R*[*F*^2^ > 2*σ*(*F*^2^)], *wR*(*F*^2^), *S*	0.038, 0.092, 1.06
No. of parameters	386
No. of restraints	0
H-atom treatment	H atoms treated by a mixture of independent and constrained refinement
Δρ_max_, Δρ_min_ (e Å^−3^)	0.60, −0.52

**Table 2 ijms-24-01745-t002:** Selected bond lengths in [Cu^II^_2_(Tolf)_4_(MeOH)_2_]∙2MeOH.

Bond	Bond Length [Å]
Cu–O1A ^i^	1.9540(13)
Cu–O2A	1.9593(13)
Cu–O1B ^i^	1.9689(15)
Cu–O2B	1.9692(14)
Cu···Cu ^i^	2.5880(4)
Cu–O1M	2.1958(13)
O1A–C1A	1.269(2)
O2A–C1A	1.276(2)
C1A–C2A	1.487(2)
C2A–C3A	1.423(2)
C3A–N1A	1.380(2)
N1A–C8A	1.407(2)
C9A–C14A	1.512(2)
C10A–Cl1	1.7511(18)
O1B–C1B	1.268(2)
O2B–C1B	1.277(2)
C1B–C2B	1.487(2)
C2B–C3B	1.423(2)
N1B–C3B	1.384(2)
N1B–C8B	1.413(2)
C9B–C14B	1.509(3)
C10B–Cl2	1.754(2)

Symmetry code: i = −x + 1, −y + 1, −z + 1.

**Table 3 ijms-24-01745-t003:** Selected angles in [Cu^II^_2_(Tolf)_4_(MeOH)_2_]∙2MeOH.

Angle	Angle Value [^o^]
O1A–Cu–O2A ^i^	169.91(5)
O1A–Cu–O2B ^i^	91.27(6)
O2A–Cu–O1B ^i^	88.57(6)
O2A–Cu–O2B	88.63(6)
O1B–Cu–O2B ^i^	169.77(5)
O1A–Cu–O1M ^i^	95.45(6)
O2A–Cu–O1M	94.58(6)
O1B–Cu–O1M ^i^	93.05(6)
O2B–Cu–O1M	96.99(6)
O1A–C1A–O2A	123.83(15)
O1A–C1A–C2A	117.40(15)
O2A–C1A–C2A	118.77(15)
O1B–C1B–O2B	123.99(16)
O1B–C1B–C2B	117.00(16)
O2B–C1B–C2B	119.01(15)
C3A–N1A–C8A	127.81(15)
C3B–N1B–C8B	126.21(16)

Symmetry code: i = −x + 1, −y + 1, −z + 1.

**Table 4 ijms-24-01745-t004:** Geometry of hydrogen bonds in [Cu^II^_2_(Tolf)_4_(MeOH)_2_]∙2MeOH.

D–H∙∙∙*A*	D–H [Å]	H∙∙∙*A* [Å]	D∙∙∙*A* [Å]	D–H∙∙∙*A* [°]
N1A–H1A∙∙∙O2A	0.80(2)	2.00(2)	2.654(2)	139(2)
N1B–H1B∙∙∙O2B	0.91(2)	1.97(2)	2.669(2)	133(2)
O1M–H1M∙∙∙O2M	0.74(3)	1.94(3)	2.682(2)	174(3)
O2M–H2M∙∙∙O1B ^i^	0.83(3)	2.52(3)	3.008(2)	119(3)

Symmetry code: i = −x + 1, −y + 1, −z + 1.

**Table 5 ijms-24-01745-t005:** QTAIM parameters for the chemical bonds and intermolecular interactions for [Cu^II^_2_(Tolf)_4_(MeOH)_2_]∙2MeOH, QEBQIX, and ACURCU01.

Bonds	ρ(r)	∇^2^(r)	ε(r)	d[Å]	δ(A,B)
[Cu^II^_2_(Tolf)_4_(MeOH)_2_]∙2MeOH					
Cu∙∙∙Cu ^i^	0.0319	0.0660	0.0096	0.0000	0.4990
Cu–O1A ^i^	0.0858	0.4508	0.0170	0.0017	0.4161
Cu–O1B ^i^	0.0827	0.4289	0.0188	0.0017	0.4010
Cu–O2A	0.0854	0.4442	0.0170	0.0013	0.4082
Cu–O2B	0.0832	0.4253	0.0151	0.0014	0.4074
O1A–C1A	0.3609	−0.3981	0.0342	0.0006	1.0692
O2A–C1A	0.3561	−0.4321	0.0367	0.0005	1.0673
O1B–C1B	0.3626	−0.4052	0.0352	0.0005	1.0691
O2B–C1B	0.3550	−0.4374	0.0360	0.0005	1.0666
C1B–C2B	0.2644	−0.6569	0.1389	0.0001	0.9734
C1A–C2A	0.2647	−0.6595	0.1374	0.0001	0.9723
Cu–O1M	0.0498	0.2201	0.0531	0.0011	0.2520
Cl2∙∙∙C2B ^ii^	0.0053	0.0159	4.8375	0.1059	0.0261
Cl1–C10A	0.1956	−0.2760	0.0609	0.0000	1.1066
Cl2–C10B	0.1956	−0.2760	0.0609	0.0000	1.1066
H11B∙∙∙C4A ^ii^	0.1956	0.0092	0.6618	0.0183	0.0097
Symmetry codes: i = −x + 1, −y + 1, −z + 1; ii = −x + 1, −y, −z + 2					
QEBQIX					
Cu∙∙∙Cu ^ii^	0.0260	0.0591	0.0176	0.0001	0.4169
Cu–O2	0.0557	0.2747	0.0215	0.0021	0.2764
Cu–O3	0.0761	0.3825	0.0224	0.0020	0.3792
Cu–O4	0.0793	0.4184	0.0237	0.0023	0.3819
Cu–O5	0.0806	0.4089	0.0186	0.0014	0.4048
Cu^ii^–O6	0.0822	0.4201	0.0170	0.0011	0.4085
O3–C6	0.3763	−0.3093	0.0384	0.0005	1.0948
O4–C7	0.3718	−0.3336	0.0372	0.0005	1.0817
O5–C6	0.3753	−0.3278	0.0402	0.0005	1.0965
O6–C7	0.3724	−0.3837	0.0482	0.0004	1.1089
C6–C8	0.2468	−0.5667	0.0653	0.0001	0.8162
C7–C9	0.2544	−0.6160	0.0704	0.0001	0.8590
Symmetry code: ii = −x, −y, −z					
ACURCU01					
Cu∙∙∙Cu ^ii^	0.0298	0.0633	0.0114	0.0000	0.4315
Cu–O(1)	0.0791	0.4095	0.0132	0.0018	0.3891
Cu–O(2)	0.0830	0.4292	0.0238	0.0014	0.4018
Cu–O(3)	0.0843	0.4445	0.0179	0.0018	0.4096
Cu–O(4)	0.0822	0.4184	0.0220	0.0014	0.4022
Cu–O(5)	0.0545	0.2642	0.0144	0.0026	0.2742
O(2)–C(1)	0.3640	−0.4092	0.0347	0.0005	1.0739
C(1)–O(3) ^ii^	0.3814	−0.2553	0.0409	0.0005	1.0954
O(1)–C(3)	0.3717	−0.2895	0.0229	0.0005	1.0620
C(3)–O(4) ^ii^	0.3742	−0.3495	0.0428	0.0005	1.1049
C(1)–C(2)	0.2544	−0.6161	0.0602	0.0002	
C(3)–C(4)	0.2532	−0.6048	0.0747	0.0001	
Symmetry code: ii = −x, −y, −z					

Symbols: ρ(r)—electron density at the BCP (bond critical point); ∇^2^(r)—Laplacian electron density; ε(r)—ellipticity of the electron density; d[Å]—the difference between the length of the bond path and the distance between the atoms linked by this bond.

**Table 6 ijms-24-01745-t006:** QTAIM parameters for the selected atomic basins for [Cu^II^_2_(Tolf)_4_(MeOH)_2_]∙2MeOH, QEBQIX, and ACURCU01.

Atom	q(A)	N(A)	%δ(A,A’)	δBond(A,A’)/2
[Cu^II^_2_(Tolf)_4_(MeOH)_2_]∙2MeOH				
Cu	1.0854	27.9146	4.96380	1.1623
Cu ^i^	1.0968	27.9032	4.97900	1.1641
O1B	−1.1178	9.1178	12.6534	0.7642
O1A	−1.1264	9.1264	12.5390	0.7426
O2B	−1.1237	9.1237	12.4814	0.7387
O2A	−1.1224	9.1224	12.6440	0.7627
C1B	1.5054	4.4946	37.0094	1.5546
C1A	1.5018	4.4982	36.9938	1.5544
Symmetry code: i = −x + 1, −y + 1, −z + 1				
QEBQIX				
Cu	1.1063	27.8937	4.8781	1.1339
Cu ^ii^	1.1063	27.8937	4.8779	1.1338
O3	−1.1427	9.1427	12.1171	0.7370
O4	−1.1575	9.1575	12.2690	0.7450
O5	−1.1227	9.1227	12.2236	0.7507
O6	−1.1062	9.1062	12.2468	0.7586
C6	1.5921	4.4079	36.8333	1.5038
C7	1.5770	4.4230	36.9083	1.5248
O1	−1.2126	9.2126	11.4775	0.7210
Symmetry code: ii = −x, −y, −z				
ACURCU01				
Cu	1.0937	27.9063	4.9487	1.1542
Cu ^iii^	1.0950	27.9050	4.9512	1.1570
O(1)	−1.1702	9.1702	12.2270	
O(2)	−1.1321	9.1321	12.3285	
O(3)	−1.1456	9.1456	12.1876	0.7502
O(4)	−1.1082	9.1082	12.2871	0.7543
O(5)	−1.1643	9.1643	11.0018	
O(5) ^iii^	−1.1560	9.1560	11.0123	
Symmetry code: iii = −x + 1, −y + 1, −z + 1				

## Data Availability

Structural data are available in the Cambridge Structural Database with deposition number: 2202870. Calculation data can be obtained from the authors upon request.

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
