# Peer review of "Crystal Structure and Chemical Bonds in [CuII2(Tolf)4(MeOH)2]∙2MeOH"

_ijms, 2023, doi:10.3390/ijms24021745_

Round 1

Reviewer 1 Report (New Reviewer)

The work of Irena Majerz et al. considers one example of copper bidentate complexes based on tolfenamate ligand.
A description of the structure of the obtained complex in the crystalline state forms the major part of the work. However, in the introduction it is mentioned about the ability of metal ions to increase the solubility of some organic substances in physiological media. However, this aspect is completely absent from the article. I can assume that such copper(II) complexes are poorly soluble in water or in aqueous media. Therefore, the question arises for what this study may be important? On the other hand, if we are talking about physiological media, the stability of the complex is an important factor. The complex contains methanol in its composition, which is also not promising for biological studies. I would recommend considering a number of solvents (DMSO in the first turn) to replace methanol. Thus, the introduction and objectives are not fully consistent with the content. This paper is more appropriate for the journal Inorganics.

Author Response

Reviewer 1

The work of Irena Majerz et al. considers one example of copper bidentate complexes based on tolfenamate ligand.
A description of the structure of the obtained complex in the crystalline state forms the major part of the work. However, in the introduction it is mentioned about the ability of metal ions to increase the solubility of some organic substances in physiological media. However, this aspect is completely absent from the article. I can assume that such copper(II) complexes are poorly soluble in water or in aqueous media. Therefore, the question arises for what this study may be important? On the other hand, if we are talking about physiological media, the stability of the complex is an important factor. The complex contains methanol in its composition, which is also not promising for biological studies. I would recommend considering a number of solvents (DMSO in the first turn) to replace methanol. Thus, the introduction and objectives are not fully consistent with the content. This paper is more appropriate for the journal Inorganics.

An important goal of receiving coordination compounds of drug is to improve the properties of the drug itself. The presented coordination dinuclear copper(II) compound is a model compound in terms of structure, although it exhibits similar solubility as tolfenamic acid. However, the main aim of the work was to investigate the interactions between the Cu2+ ions as a vital for this class of compounds. Crystals of adequate quality for X-ray examinations were obtained only from methanol solution.

It may be expected that similar interactions stabilizing the crystal will be formed when other solvents, such as ethanol, water, DMSO, coordinate instead of methanol. Importantly, analysis of the structure revealed that the other weak interactions such as Cl···π and C–H···π have greater impact on the crystal packing compared to interactions involving methanol molecules. 

Reviewer 2 Report (New Reviewer)

The authors described the synthesis and X-ray crystallography of a new complex [CuII2(Tolf)4(MeOH)2]∙2MeOH. I consider that Raman and thermal analysis should be taken. In this sense, IR, Raman, TGA, and DSC data should be properly explained in the manuscript. All figures should be included in the manuscript. Thus, I recommend the publication of the manuscript after major revision. Overall, the following suggestions are included:  (1) An explanation of the spectroscopic information is absent. Thus, Raman´s data should be included. Overall, IR and Raman data should be included and explained in the manuscript. (2) An explanation of the thermal information is absent. Thus, TGA and DSC data should be included. Overall, TGA and DSC data should be included and explained in the manuscript. (3) Using the .CIF file and CrystalExplorer program are possible to calculate HOMO and LUMO energies. Thus, the figure and its explanation should be included in the manuscript. (4) Using the HOMO and LUMO orbital energies is possible to determine global reactivity descriptors. Thus, the figure and its explanation should be included in the manuscript.

Author Response

Reviewer 2

The authors described the synthesis and X-ray crystallography of a new complex [CuII2(Tolf)4(MeOH)2]∙2MeOH. I consider that Raman and thermal analysis should be taken. In this sense, IR, Raman, TGA, and DSC data should be properly explained in the manuscript. All figures should be included in the manuscript. Thus, I recommend the publication of the manuscript after major revision. Overall, the following suggestions are included:  (1) An explanation of the spectroscopic information is absent. Thus, Raman´s data should be included. Overall, IR and Raman data should be included and explained in the manuscript. (2) An explanation of the thermal information is absent. Thus, TGA and DSC data should be included. Overall, TGA and DSC data should be included and explained in the manuscript. (3) Using the .CIF file and CrystalExplorer program are possible to calculate HOMO and LUMO energies. Thus, the figure and its explanation should be included in the manuscript. (4) Using the HOMO and LUMO orbital energies is possible to determine global reactivity descriptors. Thus, the figure and its explanation should be included in the manuscript.

According to the Reviewer guidelines IR, Raman, TGA, and DSC have been analyzed in the manuscript in the 3.5. Materials and synthetic procedures:

Raman (cm−1 ): 3160 w, 3130 w, 3078 w, 1980 w, 1951 w, 1915 w, 1858 w, 1611 vs, 1580 s, 1461 w, 1384 s, 1319 w, 1280 s, 1226 w, 1204 w, 1158 s, 1075 m, 1047 m, 849 s, 685 vw, 616 m, 520 m.

Ihe IR spectrum exhibits the weak bands at 3330 (w) cm-1 derived from ν(NH) and at 3078 (vw) cm-1 attributed to ν(CH) from aromatic ring. Bands at 2952 (vw) and 2833 (vw) cm-1 can be assigned to νasym(CH3) and νsym(CH3), respectively.
Infrared spectrum for the studied crystal exhibit characteristic bands observed at 1622 (m) and 1392 (vs) cm-1 that can be assigned to νasym(COO-) and νsym(COO-) according to Hurtado et. al. [70] [the band at 1392 (vs) cm-1 is broad, therefore a composition of vibrations of νsym(COO-) and δas(CH3) [71] can occur]. Bands at 1586 (s), 1565 (m) and 1512 (s) cm-1 can be attributed to a composition of ν(CN), bending vibrations of δ(NH) [71] and stretching vibrations of aromatic rings of ν(CC) [72], which correspond to analogous vibrations observed in the tolfenamic acid. The bands at 1459 (s) and 1447 (s) cm-1 could be assigned to δ(CH) as well as ν(CC) from aromatic rings. In the region of 1300–1000 cm-1 the δ(CH) from aromatic ring appear, but the 1286 (s) cm-1 can also be assigned to ν(CN) vibration. According to Jabeen et. al. [71] the strong band observed at 1014 (s) cm-1 can be attributed to ν(CCl) as well as ρ(CH3). In the region of 1000–675 bands observed can be assigned to δ(CH). Bands observed between 540–400 cm-1 may be interpreted as derived from ν(CCl) and δ(CCl) [71,72].

In the Raman spectrum bands at about 3160, 3130, 3078 cm-1 could be assigned to ν(CH) from aromatic rings. Broad weak bands with maximum at about 2900 cm-1  can be attributed to νasym(CH3) and νsym(CH3) vibrations. Bands at about 1611 (vs) and 1580 (s) cm-1 can be assigned to δ(NH) and ν(CN) respectively. At 1461 (w) cm-1 the deformation modes δ(CH) can be expected. The strong band at 1384 (s) cm-1 can be interpreted as derived from δ(CH3). The bands observed at 1319 (w), 1280 (s) cm-1 can be attributed to ν(CN) and at 1226 (w), 1204 (w), 1158 (s), 1075 (m), 1047 (m), 849 (s) cm-1 can be associated with ring deformations δ(CH) and δ(CC). At 685 (vw) cm-1 the deformation modes δ(NH) can be assigned. Bands observed at about 616 (m), 520 (m) cm-1 may be interpreted as derived from δ(CCl) [71].

Equipment information has been added:

3.2. DTA, DTG

DTA (Differential Thermal Analysis) and DTG (Differential Thermal Gravimetry) were carried out by means of a Seteram SETSYS 16/18 instrument in the temperature range 330–870 K on heating run at the rate of 5 K/min under N2.

3.2. IR, Raman

IR studies in the range of 4000 – 400 cm-1 were carried out using Thermo Scientific USA model Nicolet iS50 Fourier Transform Infrared Spectrometer FTIR using ATR. Raman spectrum was collected using Nicolet iS50 Raman.

The authors suggest that it would be better to include IR and Raman spectra as well as TGA and DSC diagrams in the supplementary materials, because these studies, although very important, complement the article, which is extensive and already contains 6 tables and 7 figures. 

The HOMO and LUMO, orbitals, electrostatic potential and Fukui indices have been calculated and the analysis is included in the manuscript in 2.4.

Round 2

Reviewer 1 Report (New Reviewer)

In the manuscript of Majerz and others the case of the formation of a binuclear copper(II) complex is described in sufficient detail. Structural data are presented as well as calculations demonstrating the contact between the two copper ions. As a major comment to this paper remains, the lack of correlation of the presented study with the supposed biological activity of this complex.

This manuscript is a resubmission of an earlier submission. The following is a list of the peer review reports and author responses from that submission.

Round 1

Reviewer 1 Report

This paper describes Cu-Cu bonds, Cu-O bonds based on X-ray crystallography of Paddle-Wheel-bis-Cu(II) complex with Tolfemate ligands by Majerz and Krawczy. Many papers have been published on Paddle-Wheel-bis-Cu(II) complexes, but this paper only discusses one Paddle-Wheel-bis-Cu(II) complex. Furthermore, Paddle-Wheel-bis-Cu(II)-DMF and Paddle-Wheel-bis-Cu(II)-H2O complexes were reported in Ref. 38. This paper is not considered novel.

The authors claim to have extracted 1,344 Paddle-Wheel-bis-Cu(II) complexes from the CSD database, but a similar extraction by the reviewers identified 1,570 data. Various conditions can be attached to data retrieval, but this paper does not explain the conditional retrieval of CSD. Therefore, I could not be verified. The authors compared the Cu-Cu distances of five reported Paddle-Wheel-bis-Cu(II) complexes with Tolfemate and Femate ligands (ARALUA, EDUBOX, POMHOP, EDUBUD, MADTES). 15 data of complexes were deposited in the CSD database. The average Cu-Cu distance of these is 2.607A±0.056, which is slightly shorter than the Cu-Cu distance of the complex reported in this paper.

If the above points are to be fully discussed, it should be compared with the average values of the complexes deposited in the CSD database. A detailed comparative study with CSD data is required for acceptance of this paper. Alternatively, it should be submitted to another specialized journal.

Minor issues

Line 26 determinates -> determinants

Line 26 method -> methods or ways

Lines 179 and 227 bond -> bonds

Line 272 other -> the other

Line 277 participate two -> participate in two

Line 277 percentage -> percentages

Line 279 interactions -> interaction

Line 279 cation -> cations

Line 301 component -> components

All refs. Check format!

There is no Ref. 37 in the manuscript. In addition, refs. 37 and 56 are the same.

Check ref. 63.

Author Response

Reviewer 1

This paper describes Cu-Cu bonds, Cu-O bonds based on X-ray crystallography of Paddle-Wheel-bis-Cu(II) complex with Tolfemate ligands by Majerz and Krawczyk. Many papers have been published on Paddle-Wheel-bis-Cu(II) complexes, but this paper only discusses one Paddle-Wheel-bis-Cu(II) complex. Furthermore, Paddle-Wheel-bis-Cu(II)-DMF and Paddle-Wheel-bis-Cu(II)-H2O complexes were reported in Ref. 38. This paper is not considered novel.

The Paddle-Wheel complexes are known. The novelty of this work is a detailed analysis of the bond between both Cu2+ ions. The abstract and the introduction paragraph have been changed to stress the novelty of this paper.

The authors claim to have extracted 1,344 Paddle-Wheel-bis-Cu(II) complexes from the CSD database, but a similar extraction by the reviewers identified 1,570 data. Various conditions can be attached to data retrieval, but this paper does not explain the conditional retrieval of CSD. Therefore, I could not be verified.

The number of hits in the CSD database depends on the distance between ions used in the search. The previous criterium adopted was the Cu(II)∙∙∙Cu(II) distance in the range of the sum of van der Waals radii, where the value of the van der Waals radius for Cu was determined by Bondi (Bondi, A. "van der Waals Volumes and Radii". J. Phys. Chem. 1964, 68 (3) 441–451. doi:10.1021/j100785a001.)

Authors corrected sentences in paragraph 2. Corrections are marked by yellow:

To date, the Cambridge Structural Database [27] presents 1613 hits corresponding to the search for paddle-wheel-like double-core Cu(II) deposited structures with carboxylate ligands. The structures were searched according to the following criteria: the range for Cu(II)∙∙∙Cu(II) distance: 2.40 to 3.27 Å. Among them 778 hits are the structures of compounds, in which, in addition to carboxylates, there are also other ligands coordinating to copper(II) by oxygen atom, while in 672 hits non-carboxylate ligands are linked to Cu(II) via nitrogen atom.

The authors compared the Cu-Cu distances of five reported Paddle-Wheel-bis-Cu(II) complexes with Tolfemate and Femate ligands (ARALUA, EDUBOX, POMHOP, EDUBUD, MADTES). 15 data of complexes were deposited in the CSD database. The average Cu-Cu distance of these is 2.607A±0.056, which is slightly shorter than the Cu-Cu distance of the complex reported in this paper. If the above points are to be fully discussed, it should be compared with the average values of the complexes deposited in the CSD database. A detailed comparative study with CSD data is required for acceptance of this paper. Alternatively, it should be submitted to another specialized journal.

The analyzed compound is a new coordination compound that belongs to the known group of paddle-wheel-like structures. However, the most important in the presented work is the discussion about the nature of Cu(II)...Cu(II) bonds. We carried out the analysis for the chemical compound we obtained and two others selected from the database. Although there are some theoretical studies in the literature that use the QTAIM method, they do not fully describe the nature of the bonds and apply to copper(I) compounds. In this work, we do not investigate the critical points, but the properties of the bonds. So far, the metalmetal interactions in multinuclear coordination compounds have not been studied by theoretical methods, treating them as non-covalent interactions.

Minor issues

Line 26 determinates -> determinants

Line 26 method -> methods or ways

Lines 179 and 227 bond -> bonds

Line 272 other -> the other

Line 277 participate two -> participate in two

Line 277 percentage -> percentages

Line 279 interactions -> interaction

Line 279 cation -> cations

Line 301 component -> components

All refs. Check format!

There is no Ref. 37 in the manuscript. In addition, refs. 37 and 56 are the same.

Check ref. 63.

The corrections have been introduced. The references have been checked and corrected.

Reviewer 2 Report

The abstract is not good enough, poor quality. I don't really see the novelty in this work. The authors should specify  more clearly the novelty. Why this work is an advance in the topic? it is not clear at all.

Author Response

The abstract is not good enough, poor quality. I don't really see the novelty in this work. The authors should specify  more clearly the novelty. Why this work is an advance in the topic? it is not clear at all.

The abstract and introduction paragraphs have been completed. 

Reviewer 3 Report

This study concerns the analysis of the X-ray structure of a dinuclear copper(II) complex with tolfenamic acid and theoretical study of its chemical bonds. Although the study is well structured and gives a nice overview of different computational techniques to assess the nature of the Cu2+ Cu2+ interaction, I would not recommend its publication in IJMS. The paper lacks of novelty since several similar compounds were previously obtained and the studied compound does not present significant differences from them. Also, the final findings are not so relevant in my opinion since the interaction between copper cations is barely detected. I would suggest pubblication in a more specific journal (e.g. Crystals).

Other points are:

- The analysis of some other chemical bond beside the bimetallic interaction (e.g. C-O, C-C covalent bonds) seems unnecessary.

- In the introduction the use of metal complexes to increase the bioavailability of poorly water soluble molecules is cited, I think some citations concerning this technique would be useful

- is [CuII2(Tolf)4(MeOH)2]∙2MeOH a new compound? If so, a synthetic procedure, yield and full characterization should be required. The X-ray structure of a single crystal may not account for the whole compound.

Author Response

This study concerns the analysis of the X-ray structure of a dinuclear copper(II) complex with tolfenamic acid and theoretical study of its chemical bonds. Although the study is well structured and gives a nice overview of different computational techniques to assess the nature of the Cu2+ Cu2+ interaction, I would not recommend its publication in IJMS. The paper lacks of novelty since several similar compounds were previously obtained and the studied compound does not present significant differences from them. Also, the final findings are not so relevant in my opinion since the interaction between copper cations is barely detected. I would suggest pubblication in a more specific journal (e.g. Crystals).

Other points are:

- The analysis of some other chemical bond beside the bimetallic interaction (e.g. C-O, C-C covalent bonds) seems unnecessary.

The authors claim that investigating the nature of other chemical bonds is also important in comparing the results and illustrate strength of the Cu2+…Cu2+ interaction.

- In the introduction the use of metal complexes to increase the bioavailability of poorly water soluble molecules is cited, I think some citations concerning this technique would be useful

The references: 12 -14 have been added

- is [CuII2(Tolf)4(MeOH)2]∙2MeOH a new compound? If so, a synthetic procedure, yield and full characterization should be required. The X-ray structure of a single crystal may not account for the whole compound.

The paragraph 3.3. Materials and synthetic procedures has been completed.

Round 2

Reviewer 1 Report

This revised paper does not appear to be adequately revised for Reviewer 1's observations. Especially CSD database 2022.1.0, the latest database is CSD ver. 5.43. Since the data are used in the paper, a clear rationale is required. Although Reviewer 1 did not point it out, perhaps another reviewer requested and added the spectral data of the Cu(II) complex. An NMR of the Cu(II) complex has been described, but copper(II) is paramagnetic. Therefore, it is not possible to understand what the data in which NMR is described mean.

Reviewer 2 Report

the authors did not address my concerns about the lack of novelty of the paper. Therefore, I don't consider it good enough for publication in this journal

Reviewer 3 Report

I do appreciate the changes the authors made on the manuscript, that improved the understanding of the aim of this paper (except for the synthetic procedure for the copper complex, which is approximate and inaccurate in several parts, e.g. mM instead of mmol).
Still, I don't change my mind and I think the overall quality is not enough for IJMS for the reasons I stated in the previous review.

The comparison of the strength of the Cu2+ Cu2+ interactions with the one of an hydrogen bond is an interesting point, however its contribute doesn't seem so relevant, since, as the authors state, the stabilising force of the dinuclear complex is the Cu-carboxylates interaction. So, the sentence "this interaction can be considered as one of the non-covalent interactions responsible for the arrangement of molecules in the crystal and thus determining the physicochemical properties"  sounds a bit pretentious to me, if based only on theoretical calculations on the solid state structure.

the word "pnictogen" is mispelled